# Diet Affects the Temperature–Size Relationship in the Blowfly *Aldrichina grahami*

**DOI:** 10.3390/insects15040246

**Published:** 2024-04-03

**Authors:** Guanjie Yan, Dandan Li, Guangshuai Wang, Lingbing Wu

**Affiliations:** 1Henan International Joint Laboratory of Insect Biology, Henan Key Laboratory of Insect Biology in Funiu Mountain, Nanyang Normal University, Nanyang 473061, China; guanjiey@gmail.com (G.Y.);; 2Farmland Irrigation Research Institute, Chinese Academy of Agriculture Sciences/Shangqiu Station of National Field Agro-Ecosystem Experimental Network, Shangqiu 476000, China; 3School of Tropical Agriculture and Forestry, Hainan University, Renmin Road 58, Haikou 570228, China

**Keywords:** body size, ectotherms, growth curve, temperature–size rule

## Abstract

**Simple Summary:**

In warmer environments, the growth of ectotherms is usually accelerated and is expected to result in maturation at a larger body size. Yet, most ectotherms exhibit a plastic reduction in body size (the temperature–size rule, TSR), which has caused people confusion. To explore these mechanisms, we reared *Aldrichina grahami* at 20 °C, 25 °C, and 30 °C, and added a nutritional challenge by using dilutions of pork liver paste to provide diets that ranged in quality from high (undiluted) to moderate (1/8), low (1/16), and poor (1/24). The growth of larvae was measured, and growth curves were fitted to the relationships between growth rate and weight for the third instar larvae. Our results showed that when the TSR was followed as the temperature increased, there was a cross-over point that divided the two growth curves into early and later stages, which could be used to help understand the life-history puzzle in warmer temperatures, with the instantaneous growth rate being faster in the early stages of development and then slower in later stages. This study reminds us that animals have evolved to cope with multiple simultaneous environmental changes, and it has thus offered a better understanding of life-history puzzles.

**Abstract:**

In warmer environments, most ectotherms exhibit a plastic reduction in body size (the temperature–size rule, TSR). However, in such environments, growth is usually accelerated and would be expected to result in maturation at a larger body size, leading to increases in fecundity, survival, and mating success, compared to maturation at a smaller size (the ‘life-history puzzle’). To explore these mechanisms, we reared *Aldrichina grahami* at 20 °C, 25 °C, and 30 °C, and added a nutritional challenge by using dilutions of pork liver paste to provide diets that ranged in quality from high (undiluted) to moderate (1/8), low (1/16), and poor (1/24). Larvae were randomly sampled for weighing from hatching. Growth curves were fitted to the relationships between growth rate and weight for the third instar larvae. Our results showed that body size was affected by an interaction between temperature and diet, and that following or not following the TSR can vary depending on underfeeding. Moreover, when the TSR was followed as temperature increased, there was a cross-over point that divided the two growth curves into early and later stages, which could be used to help understand the life-history puzzle in warmer temperatures, with the instantaneous growth rate being faster in the early stages of development and then slower in later stages. This study reminds us that animals have evolved to cope with multiple simultaneous environmental changes, and it has thus offered a better understanding of life-history puzzles.

## 1. Introduction

Body size is one of the most important organismal traits due to its strong links to anatomy, physiology, behavior, ecology, and life history [1,2,3,4]. The approximate size of each species is genetically pre-determined but also varies in response to the developmental environment [5,6,7,8]. Temperature is one of the most important environmental factors because it has a marked effect on body size—ectotherms grow faster but mature at a smaller body size in warmer environments. Despite a few exceptions [9], this phenomenon is so general that it has become universally known as the ‘temperature–size rule’ (TSR) [10,11,12,13,14]. Similarly, individuals from natural populations in colder climates are typically larger than individuals from populations in warmer environments, a phenomenon known as ‘Bergmann’s rule’ [15,16,17]. It is essential that we deepen our understanding of the mechanisms that underpin these rules because human-induced global warming is altering temperature regimes, potentially changing the body size of organisms, and thus affecting ecological and evolutionary processes worldwide [18,19].

In addition to temperature, nutrition is an environmental factor that can have marked effects on the body size of ectotherms—for example, a high-quality diet commonly leads to a greater growth rate and a larger body size [20,21,22]. However, the effects of nutrition and temperature on body size differ because temperature mainly affects physiological rates [23], whereas diet quality mostly affects energy uptake [24,25]. Thus, high temperature usually increases growth rate [26] whereas a low-quality diet usually decreases growth rate [22], yet both situations result in a smaller body size. In addition, in ectotherms nutrition and temperature interact to affect growth rate and body size, with the slope of the body size–temperature relationship depending on the amounts of macronutrients ingested and absorbed, and the proportion allocated to growth [27,28]. A high-quality diet increases the thermal sensitivity of the growth rate in *Pieris rapae* (Lepidoptera: Pieridae), and a low-quality diet reverses the TSR in *Manduca sexta* (Lepidoptera: Sphingidae) [20,29]. The fact that the TSR could be reversed by manipulation of the diet led us to test whether a disobeyed TSR is also reversed when diet is introduced as a factor.

In addition, the different effects of nutrition and temperature on growth rate and body size present a life-history puzzle because an increase in growth rate would be expected to lead to maturity at a larger body size (high resource availability), leading in turn to increases in fecundity, survival, and mating success, in contrast to the smaller size at higher temperatures that results from the TSR [11,25,30]. Moreover, the growth curve of insect larvae is commonly sigmoidal until development reaches the end of the final instar [31], suggesting that growth rate is related to the development stage. In addition, the temperature–size relationship changes with the development stage, being weak or even positive in early stages but stronger and negative in late stages [20,32,33,34,35,36]. Given these observations, if an ectotherm species follows the TSR and shows a decrease in final body size in response to warmer temperatures, it might grow more rapidly in early development stages but not in late stages [37].

The critical challenge is distinguishing the earlier and later stages of development. In the cladoceran, for example, *Daphnia magna* (Crustacea: Cladocera), growth is constrained in large but not in small individuals of [24] and body mass affects energy acquisition and energy costs, so the early and late stages of development might be defined simply on the basis of weight [38,39]. Thus, the second aim was to test whether the life-history puzzle in the TSR could be settled by dividing larval development into early and late stages; specifically, we hypothesized that growth rate and body size would be enhanced during the initial stages under warmer temperatures but reduced during later stages.

*Aldrichina grahami* (Aldrich) (Diptera: Calliphoridae) is an important fly species that is used as an indicator for the estimation of the postmortem interval in forensic science [40]. Using a factorial design (temperature × nutrition) for rearing conditions, we used *A. grahami* to test whether (a) temperature and diet quality interact to affect larval and adult body size, and whether manipulating diet quality can alter the temperature–body size relationship; (b) the development of larvae could be divided into early and late stages at a special weight, and thus resolve the life-history puzzle in the TSR.

## 2. Materials and Methods

### 2.1. Fly Stocks and Rearing

The colony of *A. grahami* was started with 18 adults (15 females and 3 males) obtained with three fly traps baited with pork liver in Xinxiang (35°09′ N, 113°45′ E), China, in May 2020. They were identified using the key from Fan [41]. The flies were transferred to a cage (35 × 35 × 35 cm) and reared under 12:12 (L:D) h photoperiods at 30–40% relative humidity and 23–25 °C room temperature, and offered water and pork liver paste (replaced at 10.00 h daily). Eggs observed on the liver paste during replacement were collected and used for rearing the F1 generation.

To rear the F1 generation, in each cage, about 300 eggs were transferred to pork liver paste (20 g) on plastic trays (15 × 15 cm) that were placed on dry sand (about 5 cm deep) in a plastic box (30 × 25× 15 cm). Mesh was used to cover the box to prevent the larvae from escaping as well as to prevent entry by flies in the rearing room. The plastic boxes were kept in an incubator (25 ± 0.5 °C) and liver paste (100–200 g) was added at 9 am every morning until all larvae had reached the wandering stage (they climbed out of their plastic boxes and searched for a place to pupate). When all larvae had pupated, they were incubated for another 4 days, after which the pupae were removed into a plastic container (9 cm i.d. × 7 cm height) and covered with about 2 cm dry sand. The plastic container was placed in a cage (35 × 35 × 35 cm) that was maintained under 12:12 (L:D) h photoperiods at 30–40% relative humidity and 23–25 °C room temperature. From eclosion, the flies were provided ad libitum with water and a mixture (1:1 in weight) of milk powder (Anchor, New Zealand) and white sugar (Yutang, China). Notably, pork liver paste was provided for a duration of 4 h per day as adults aged from 4 to 8 days to ensure their nutritional intake for gravidity. Furthermore, pork liver paste was also provided for the purpose of egg collection and larval rearing. Three cages of the F1 generation were reared, from which the eggs were collected for use in the following studies.

### 2.2. Measurement of Growth Curves

The eggs were collected from 10-day-old flies by placing a plastic tray with 10 g pork liver in their cage for 4 h. For larval rearing, all eggs (approximately 2400) were collected and pooled together, and then about 200 eggs (based on our unpublished work that the development of larvae had not been affected by densities of 50, 100, 200, or 500 per container) were allocated to each of three replicate containers containing about 50 g of paste for each of the four dietary treatments. The eggs were then incubated and the larvae reared at 25 °C. In the preliminary experiment, nutrition was manipulated by controlling the amount of diet or the diet quality. In several of the groups where the amount was manipulated, the larval period was too short to record sufficient weight data, so we chose to focus on manipulation of diet quality. Quality was controlled by diluting the pork liver with agar gel (prepared by boiling 10 g agar in 1 L water). The preliminary experiment also showed that adult body size was not significantly reduced until the liver paste was diluted 1/8 (*w*:*w* = 1:7, 12.5%), and no larva survived when it was diluted 1/32 (1:31, 3.125%). It was thus also clear that agar gel offered little nutritional value and was also non-toxic. Thus, the final dietary treatments were formed by diluting pork liver paste with agar gel paste in four ways: high quality (100% pork liver, PL); moderate quality (1:7, 12.5% PL); low quality (1:15, 6.25% PL); and poor quality (1:23, 4.17% PL). To ensure that equal amounts of liver were available in each treatment, more food was offered in the groups fed a diluted diet. Eight larvae were randomly sampled for weighing from each of the three replicates in each treatment, every eight hours from hatching until more than half of the larvae had reached the wandering stage (they climbed out of their plastic boxes and searched for a place to pupate). Larvae (*n* = 20) in the wandering stage and fresh-eclosion adults, male (n = 20) and female (n = 20), were randomly selected for weighing. Considering that larvae in the wandering stage cease feeding and undergo weight loss, immediate weighing was conducted on the initial 20 larvae emerging from the container to minimize the potential impact of sampling time. The same protocol was used at 20 °C and 30 °C. All larvae were sampled as quickly as possible and permanently removed so as to reduce the effects of handling on development.

### 2.3. Curve Fitting for Weight and Growth Rate

As with Lepidopteran species [42], *A. grahami* larvae accumulate more than 90% of their mass in the final instar, so the weights of larvae in the third instar (the final instar of *A. grahami*) were used for fitting growth curves. Four regression equations (linear, quadratic, cubic, and quartic polynomial) were used to examine the relationship between weight and age (starting from hatching), from the start of the third instar until more than half of the larvae had reached the wandering stage. Models were compared using Akaike’s information criteria (AIC). Models with the lowest AIC value were considered to provide the best fit, and all models within ΔAIC < 2 of the best model were considered to have similar support [43]. As a result, cubic polynomial (2) regression equations were selected among different groups as follows:y = ax^3^ + bx^2^ + cx + d(1)
where y is the weight of larvae, x is the time after hatching, a, b, and c are coefficients of x, and d represents the intercepts. Another equation was obtained as derivatives of y: y’ = 3ax^2^ + 2bx + c(2)
where y’ is the growth rates of larvae at x hours after hatching; a, b, and c are also coefficients as in Equation (1). Thus, for a given age, the weight of larvae in the third instar can be calculated by Equation (1), and the corresponding growth rate can be calculated by Equation (2). Then the growth rate–weight curve can be formatted by assigning continuous values of age in y and y’.

### 2.4. Statistical Analysis

Data (weight of larvae and adults) were evaluated for normality using a Q–Q plot. To determine whether the weight of adults was affected by the gender of *A. grahami*, we used three-way ANOVAs to test the effect of the interaction between temperature, diet quality, and gender, on adult weight. No significant interaction among these three factors (F_6, 456_ = 0.846, *p* = 0.535), temperature × gender (F_2, 456_ = 0.535, *p* = 0.586), or diet quality × gender (F_3, 456_ = 0.431, *p* = 0.731) was detected, so the weights of the males and females in each group were pooled for further analysis. The weight of larvae in the wandering stage and in fresh-eclosion adults, for the high-quality diet groups, were analyzed by one-way ANOVA, with temperature as the factor. The effects of the interaction of temperature and diet were assessed by a two-way analysis of variance with temperature and diet as the factors. Further one-way ANOVAs followed for each dietary treatment with temperature as the factor. The least significant difference (LSD) was used for pairwise comparisons of temperature treatments. For all analyses, *p* < 0.05 was considered to be significant. 

Curves were fitted to the relationships between growth rate and weight for the third instar larvae using R [44]. Linear, quadratic, cubic, and quartic polynomial models were evaluated using Akaike’s information criteria (AIC). 

## 3. Results

### 3.1. The TSR for A. grahami

In the groups fed the high-quality diet, the weight of wandering-stage larvae was significantly affected by the rearing temperature (Figure 1A, ANOVA, F_2, 57_ = 21.27, *p* < 0.001). The LSD values showed that the patterns in larva weight were similar for 20 °C (112.1 ± 1.2 mg) and 25 °C (114.3 ± 1.4 mg), whereas larvae were lighter for 30 °C (104.5 ± 0.7 mg) than for 25 °C (*p* < 0.001) and 20 °C (*p* < 0.001). Adult weight was also significantly affected by the rearing temperature (Figure 1B, ANOVA, F_2, 117_ = 16.084, *p* < 0.001); again, LSD values were consistent with those of the larvae.

### 3.2. The Interaction between Temperature and Diet

The results of two-way ANOVAs for the weights of larvae and adults, with temperature and diet as the two factors, are shown in Table 1. The interaction between the two factors was significant for both larvae (F_6, 228_ = 18, *p* < 0.001) and adults (F_6, 468_ = 5, *p* < 0.001), so one-way ANOVAs with temperature as the factor were used within each dietary treatment, and with diet as the factor within each temperature treatment.

As showed in Figure 1, the weights of both wandering-stage larvae and fresh-eclosion adults were significantly affected by the rearing temperature in the groups that were fed with the moderate-quality diet (ANOVA, F_2, 57_ = 58.75, *p* < 0.001 and F_2, 117_ = 13.891, *p* < 0.001), the low-quality diet (F_2, 57_ = 7.865, *p* < 0.001 and F_2, 117_ = 16.438, *p* < 0.001), and the poor-quality diet (F_2, 57_ = 16.63, *p* < 0.001 and F_2, 117_ = 43.0, *p* < 0.001). The LSD values showed that diet could affect the temperature–size relationship of larvae in the wandering stage and fresh-eclosion adults in two patterns. On the one hand, as the temperature increased from 20 °C and 25 °C to 30 °C, a transition in weight change was observed, shifting from a significant decrease in adults and larvae with high-quality diets to either no significant difference in adults with low- and poor-quality diets and larvae with a poor-quality diet (as the temperature increased from 20 °C to 30 °C), or a significant increase in larvae with a poor-quality diet as the temperature increased from 25 °C to 30 °C (Figure 1). On the other hand, as the temperature increased from 20 °C to 25 °C, there was a transition in weight change from no significant difference of adults and larvae with a high-quality diet to a significant decrease in larvae with a poor-quality diet and in adults with low- and poor-quality diets (Figure 1). 

### 3.3. Relationships between Weight and Growth Rate

Akaike’s information criteria (AIC) for linear, quadratic, cubic, and quartic polynomial equations fitted to the data for weight and age of larvae are shown in Table 2. The main principle for polynomial selection—the smaller the AIC, the better the polynomial—ensures the selection of the simplest model with the least terms. As a result, cubic polynomials were selected because they have the smallest or similar AIC (ΔAIC < 2, Table 2). The equations in Table 2 were used to plot the relationship curves between growth rate and weight shown in Figure 2.

When the TSR was followed in a warmer environment, the two weight–growth rate curves exhibited cross-over (Figure 2D–G,I–K), and the cross-over point divides the larval growth curves into earlier and later stages. Notably, the instantaneous growth rates were higher in a warmer environment during the early stage when larval weights were lighter than the cross-over point, but lower during the later stage when larval weights were heavier than the cross-over point. In contrast, when the TSR was not followed in a warmer environment, the two weight–growth rate curves exhibited either no cross-over (Figure 2A,B,H) or cross-over at the final weight (Figure 2C,L). However, when the growth rate was determined by dividing the weight of wandering larvae by their development time, a significant increase in growth rate was observed at higher temperature (Figure 3).

## 4. Discussion

### 4.1. The Effects of Temperature on Body Size

Our results showed that, with the high-quality diet, increasing the rearing temperature from 20 °C to 25 °C caused no significant increase in the weight of wandering-stage larvae or fresh-eclosion adults, whereas increasing the rearing temperature to 30 °C significantly decreased the weights. These observations with *A. grahami* are similar to those for the beetle, *Cephaloleia placida* (Coleoptera: Chrysomelidae) [45] and for *M. sexta* [32], for which there were peaks in the thermal reaction norm of body size (reversal of the TSR at very low temperatures; TSR at field-realistic temperatures). It thus seems likely that both following and not following the TSR apply to any given ectotherm subjected to such changes in rearing temperature.

### 4.2. The Potential Explanation of the Life-History Puzzle

In ectotherms, the effect of temperature on growth rate seems to be more stable than the effect on body size, as evidenced by the existence of several reports not following the TSR in a variety of genera, including butterflies *Bicyclus anynana* (Lepidoptera: Nymphalidae) and *P. rapae* [29,46]. It seems to be very rare for warmer temperatures not to increase the growth rate—perhaps we are misled by the common practice of calculating an average growth rate (increased larva weight/larval time) [47], and we should be using an instantaneous value for growth rate at the maximum weight because, to understand the effects of temperature on size, we need to incorporate interactions between time, temperature, and body size [37]. Our results showed that growth rates were greater in warmer temperatures when they were calculated as the weight of wandering larvae divided by larval development time (Figure 3); by contrast, instantaneous growth rates, obtained by derivatives of weight with age as the variable, were affected by both temperature and weight (Figure 2). For situations where the TSR was not followed, the instantaneous growth rates were increased by warmer temperatures, either throughout all (Figure 2A,B,H) or almost all of the period of development (with exceptions around the maximum weight, Figure 2C,L). For situations where the TSR was obeyed, however, there was a weight (the cross-over point) at which the growth rates under two different temperatures are equal. In addition, the cross-over point divided development into early and late stages—with warmer temperatures, instantaneous growth rates were increased in the early stage but lower in the late stage. Hoefnagel et al. [24] reported that sensitivity of growth rates to temperature differed markedly between animals of different sizes, with daily growth peaking at temperatures between 23 and 26 °C for smaller animals (1.2 mm), while peaking between 10 and 23 °C for larger animals (2.6 mm). Our observations thus offer an explanation of the TSR life-history puzzle because, in addition to a decrease in final body size, there is also a decrease in instantaneous growth rate.

### 4.3. Interconversion between Following and Not Following the TSR

Our results showed that in the groups fed with a high-quality diet, the TSR was observed, and the weights of wandering-stage larvae and fresh-eclosion adults were lighter at 30 °C than at 20 °C and 25 °C (Figure 1 and Figure 2). However, when the diet was diluted to poor-quality, the TSR being followed was reversed to not being followed, and the weights of wandering-stage larvae and fresh-eclosion adults at 30 °C were higher than or similar with at 20 °C and 25 °C (Figure 1 and Figure 2). Similarly, resource quality can reverse the TSR, which has been reported in studies of *P. rapae* [29], *M. sexta* [20], and *Tetrahymena thermophila* (Hymenostomatida: Tetrahymenidae) [48]. Moreover, our results also demonstrated that in the groups fed with a high-quality diet, by not following the TSR, it was observed that the weights of wandering-stage larvae and fresh-eclosion adults were heavier at 25 °C than at 20 °C, while the TSR not being followed was reversed to being followed when the diet was diluted to poor-quality, where the weights of wandering-stage larvae and fresh-eclosion adults at 25 °C were lighter than at 20 °C (Figure 1 and Figure 2). These results indicated that manipulation of the diet can cause not only a reversal of following the TSR but also a possible reversion of not following the TSR.

### 4.4. Limitations

Notably, in the present study, the nutritional treatments were implemented by dilution of the diet, rather than by varying the nutrient balance (e.g., adjusting the ratio of lipid to carbohydrate to protein), perhaps affecting the metabolic outcomes and changing the thermal responses in ways that are not aligned with nutrient dilution. As a forensically important fly species, *A. grahami* commonly feeds on different tissues with various nutrient balances, rather than feeding on a diet of stable concentration and nutrient balance. Moreover, many larvae fed the poor-quality diet escaped and died before the third instar, suggesting that the poor-quality diet was stressful. Thus, in further forensic studies on insect development, other factors should be considered and more attention should be paid to the effects of nutrient balance instead of dilution.

## 5. Conclusions

Overall, this study of *A. grahami* has shown that: (1) the weights of both wandering-stage larvae and fresh-eclosion adults did not significantly increase as the rearing temperature increased from 20 °C to 25 °C, but significantly decreased at 30 °C; (2) a cross-over point divides the larval growth curves for the TSR into earlier and later stages, and the growth is accelerated under a warmer temperature in the earlier stage but decelerated in the later stage; (3) temperature and diet interactions affect the weight of both wandering-stage larvae and fresh-eclosion adults, and following and not following the TSR in specimens with a high-quality diet were interconverted by underfeeding. Based on the responses of *A. grahami* larvae to nutritional and thermal constraints during rearing, following the TSR and not following the TSR were affected by an interaction between the two environmental factors. When the TSR was obeyed, there was a weight (the cross-over point in the growth curve) at which the growth rates under two different temperatures were equal. This cross-over point could be used to help understand the life-history puzzle in warmer temperatures, with the instantaneous growth rate being high in the early stages of development and then lower in later stages.

On a general level, this study reminds us that animals have evolved to cope with multiple simultaneous environmental changes in, for example, temperature and nutrition. By observing their responses to interactions among multiple factors in our experiments, we will be better able to understand and resolve life-history puzzles and temperature–size rules.

## Figures and Tables

**Figure 1 insects-15-00246-f001:**
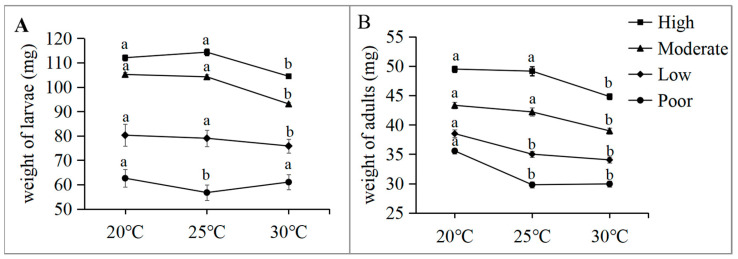
Effect of temperature on: (**A**) final larval weight and (**B**) weight of fresh-eclosion adults of *Aldrichina grahami* reared under the four dietary treatments (mean ± SEs). Different lowercase letters represent statistically significant differences (*p* < 0.05) among temperatures within each diet.

**Figure 2 insects-15-00246-f002:**
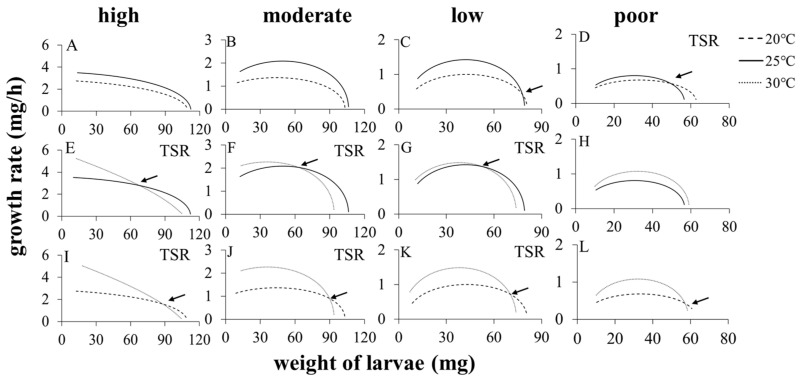
The relationships between instantaneous growth rate and weight during the rearing of *Aldrichina grahami* larvae fed high—(**A**,**E**,**I**), moderate—(**B**,**F**,**J**), low—(**C**,**G**,**K**), and poor—(**D**,**H**,**L**) quality diets, comparing 20 °C and 25 °C (**A**–**D**), 25 °C and 30 °C (**E**–**H**), and 20 °C and 30 °C (**I**–**L**). The arrow indicates the cross-over point at which the growth rate and weight curves cross at the two temperatures (i.e., growth rate was the same for both temperatures).

**Figure 3 insects-15-00246-f003:**
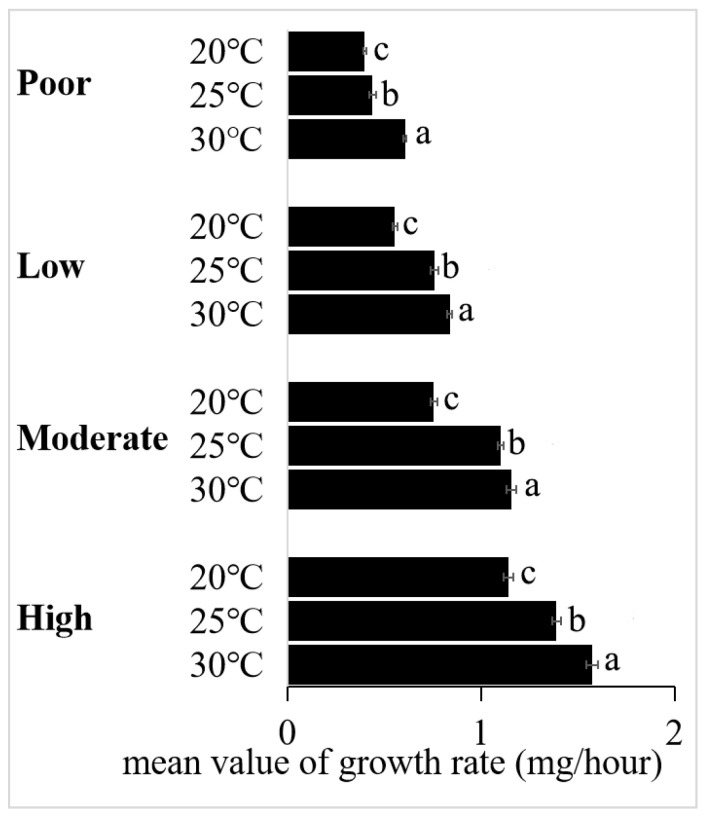
The effect of temperature on the growth rate of *Aldrichina grahami* larvae reared under four diets that differ in quality. Different lowercase letters represent statistically significant differences (*p* < 0.05) among temperatures within each diet.

**Table 1 insects-15-00246-t001:** The results of two-way ANOVAs for the weight of larvae and adults of *Aldrichina grahami* with temperature and diet as the two factors.

Developmental Stage	Factor	df	*F*	*p*
Larvae	Temperature	2, 228	58	<0.001
	Diet	3, 228	1971	<0.001
	Temperature × Diet	6, 228	18	<0.001
Adults	Temperature	2, 468	66	<0.001
	Diet	3, 468	423	<0.001
	Temperature × Diet	6, 468	5	<0.001

**Table 2 insects-15-00246-t002:** Akaike’s information criteria (AIC) and equations describing the relationships between weight and age of *Aldrichina grahami* larvae reared under four dietary treatments and three temperatures.

Diet	Temperature (°C)	AIC *				Equation	R^2^	*p*
		1	2	3	4			
high	20	465	412	413	413	y = −0.00016153 x^3^ + 0.010228 x^2^ + 2.7020 x − 101.5	0.97	<0.001
	25	421	344	342	340	y = −0.00040961 x^3^ + 0.030104 x^2^ + 2.8451 x − 98.4	0.98	<0.001
	30	352	275	272	274	y = 0.00070032 x^3^ − 0.159341 x^2^ + 12.3091 x − 217.1	0.98	<0.001
moderate	20	726	660	644	646	y = −0.00010752 x^3^ + 0.02163815 x^2^ − 0.08315955 x − 15.54660964	0.97	<0.001
	25	505	441	418	420	y = −0.00041413 x^3^ + 0.06348199 x^2^ − 1.15964988 x − 1.90000000	0.98	<0.001
	30	461	382	365	363	y = −0.00051627 x^3^ + 0.05609168 x^2^ + 0.23352667 x − 21.48125000	0.98	<0.001
low	20	644	637	619	620	y = −0.00010333 x^3^ + 0.02784374 x^2^ − 1.49937023 x + 27.83584540	0.97	<0.001
	25	440	417	400	400	y = −0.00030980 x^3^ + 0.05975379 x^2^ − 2.41729403 x + 32.89621212	0.98	<0.001
	30	428	411	380	382	y = −0.00037605 x^3^ + 0.05738587 x^2^ − 1.43509049 x + 12.19987374	0.98	<0.001
poor	20	742	739	728	729	y = −0.00005192 x^3^ + 0.01525747 x^2^ − 0.81487311 x + 15.39486690	0.94	<0.001
	25	557	545	524	525	y = −0.00011453 x^3^ + 0.02651317 x^2^ − 1.23866570 x + 20.98557831	0.96	<0.001
	30	424	421	362	360	y = −0.00025717 x^3^ + 0.04444600 x^2^ − 1.48207769 x + 19.16386364	0.99	<0.001

* Polynomial equations: 1 for linear, 2 for quadratic, 3 for cubic and 4 for quartic. Models with the lowest AIC value were considered to provide the best fit, and all models within ΔAIC < 2 of the best model were considered to have similar support. In the equations, x is age (hours) after hatching and y is weight of larva at age x.

## Data Availability

The raw data supporting the conclusions of this article will be made available by the authors on request.

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
