# Peer review of "Diet Affects the Temperature–Size Relationship in the Blowfly Aldrichina grahami"

_insects, 2024, doi:10.3390/insects15040246_

Round 1

Reviewer 1 Report

Comments and Suggestions for Authors

This paper provides valuable information about diet quality and temperature effects on development and size of a forensic relevant blow fly. This paper can be useful for areas that have high temperatures and should be investigated further. Please see below for suggestions to improve the manuscript.

Make sure to include Order and family with species presented within in text (non blow flies)

Line 92 is an incomplete sentence.

How much paste were the eggs placed on. You mention a density of 200 being sufficient place don preliminary work but you do not specify on what amount of food substrate.

Why were the first 20 wandering larvae weighed? Could that not lead to a sample error?

Should the second equation (between lines 163 & 164) by X^2?

Fig 1A & B would be better represented with 4 lines on each figure especially since the Y values are different for each of them.

You figure legend, denotes * as significantly different but you do not have * anywhere on your figure. I would clarify that the different letters signify significant differences at that P value.

Were normality tests conducted to determine if ANOVA was the most appropriate test to analyze the data?

Lines 207 – 211 state that there were significant differences in weights based on temperature for moderate, low and poor diets, but your figure indicated significant differences across temperature for all diet qualities.

Lines 213-216 – there was no significant difference between your 20 & 25 treatments in larval weight, so you stating that their weights varied doesn’t really mean much if they weren’t significantly different.

Point 2 here- that larval weights were significantly difference between 20 & 30. They were also significantly different between 25 & 30 – why is this not addressed within this point?

Point 3 does not match the figures or is written in a confusing manner. This point reads that the larval weight was higher at all diets excluding poor which is the opposite of what is shown.

What is meant by ‘familiar’ weight?

Lines 222-229- this section and the previous section are worded in a confusing manner. The difference between 20 & 25 didn’t vary, they were statistically not different from one another. These sections could be worded in a simpler manner. Something like  ‘When fed a diet of high or moderate quality there was no significant difference between adult weight at 20 and 25C with weights being significantly lower at 30C. While when provided a low or poor quality diet there was no significant difference between adult weight at 25 or 30 with

Comments on the Quality of English Language

English editing is needed for sections of this paper, grammar issues are scattered throughout the paper, and editing with strengthen the paper.

Author Response

Dear Reviewer

Thank you very much for taking the time to review this manuscript. Please find the detailed responses and the corresponding revisions/corrections in the attachment.

Regards

Reviewer 2 Report

Comments and Suggestions for Authors

The authors investigated the interaction of food and temperature, two main factors determining the rate of insect preimaginal development and the size of emerging adults, using a forensically important blow fly Aldrichina grahami as a model. The experiments were well planned and performed; the data were correctly analyzed; the results of the study can be interesting both for basic insect eco-physiology and for applied forensic entomology. Thus, the manuscript can be published, although it needs a number of corrections and improvements (see below).

Line 104: cage size 35 x 35 x 35 is not in cubic but in linear centimeters.

Line 115: Possibly, not “incubated after another 4 days” but “incubated for another 4 days”.

Line 117: cage size 35 x 35 x 35 is not in cubic but in linear centimeters.

Line 120 Was the pork liver paste offered for feeding only or also for oviposition (as in lines 105 and 124-125)? Please, clarify.

Line122: Insert “were’ between “eggs” and “collected”.

Figure 1: I guess that letters along the lines indicate pairwise significant difference between the data for different temperatures with the same diet. If yes, please, explain it clearly in the legend.

Figure 1: In the legend, it is stated that “* significantly different, P < 0.05.” but I can’t find any “*” symbol in this figure.

Figure 1: The significance of the pairwise differences between the data for different diets with the same temperature should be also indicated (e.g., by the capital letters).

Lines 202-203: Please, do also one-way ANOVA with diet as a factor within each temperature (see also the last comment to Fig. 1).

Lines 204-205: Please, indicate the insect name Aldrichina grahami in the heading of Table 1.

Line 213: Please, explain what is “the familiar larval weight”.

Line 215 etc. Please, replace everywhere “significant lower (or higher)” with much more commonly used “significantly lower (or higher)”.

Lines 219-220. Possibly, it is my misunderstanding, but this text is a bit unclear: it is stated that larval TSR “in groups fed high-, moderate- and low-quality diets were reversed, with the weight was higher at 30 °C than at 25 °C” However, in Fig. 1 it is seen that with these diets larval weight at 30°C was not higher but lower than that at 25°C. If you mean that the weight was higher at 30 °C than at 25 °C only at the poor diet, this should be clearly stated in the text.

Line 247: delete commas before and after “during rearing”.

Lines 262-264: This sentence is almost the same as that in lines 271-273. Moreover, the whole paragraph (lines 262-269) represents a short summary that I would suggest to replace to the end of the Discussion (Conclusions).

Line 264: Again, I would suggest replacing of “significant decreased” with “significantly decreased”.

Lines 289-298: I think that this text, as well as Figure 3, should be placed in the Results, not in the Discussion section of the paper.

Lines 311-312: I think that “weights..... were heavier“ is not a good English. Please, consider either “weights.... were higher” or “larvae... were heavier”.

Line 338: I would suggest not “are equal” but “were equal”.

Author Response

(The authors gave the same response as above.)

Round 2

Reviewer 1 Report

Comments and Suggestions for Authors

The edits to the manuscript have improved the manuscript overall.  I have a few comments below:

My one concern with Figure 1 it that it is a bit confusing with all the letters. I think you could remove the capital letters since none of them overlap and state in the legend that there were significant differences across diets  and therefore the analysis is between each treatment across temperatures (or something similar). If the authors and editors feel capital letters should stay please keep it.

Line 220, affects should be affect

Lines 219-227- this doesn't match what your figure is showing with how it is written.

On figure 3 it is still not clear which treatments are significantly different from one another. For example, the three bars under the poor diet, are 20 & 25 significantly different? Or is it showing that 20 & 25 are different from 30? That is unclear for each diet.

Author Response

Dear Reviewer

Thank you very much for taking the time to this manuscript, the following are responses to your comments, please take a look.

1. My one concern with Figure 1 it that it is a bit confusing with all the letters. I think you could remove the capital letters since none of them overlap and state in the legend that there were significant differences across dietsand therefore the analysis is between each treatment across temperatures (or something similar). If the authors and editors feel capital letters should stay please keep it.

Response: Agree, the capital letters and relative description in results have been removed, see new lines 200-204.

2. Line 220, affects should be affect

Response: Done, see new line 219.

3. Lines 219-227- this doesn't match what your figure is showing with how it is written.

Response: The relevant section has undergone modifications as evidenced by the revised content presented in lines 220-225.

4. On figure 3 it is still not clear which treatments are significantly different from one another. For example, the three bars under the poor diet, are 20 & 25 significantly different? Or is it showing that 20 & 25 are different from 30? That is unclear for each diet.

Response: Agree, the symbol "*" has been substituted with distinct letters to demonstrate significant difference, see new lines 262-265.

Regards

Guanjie